# VideoVerse: How Far is Your T2V Generator from a World Model?

## Abstract

The recent rapid advancement of Text-to-Video (T2V) generation technologies, which are critical to build "world models", makes the existing benchmarks increasingly insufficient to evaluate state-of-the-art T2V models. First, current evaluation dimensions, such as per-frame aesthetic quality and temporal consistency, are no longer able to differentiate state-of-the-art T2V models. Second, event-level temporal causality, which not only distinguishes video from other modalities but also constitutes a crucial component of world models, is severely underexplored in existing benchmarks. Third, existing benchmarks lack a systematic assessment of world knowledge, which are essential capabilities for building world models. To address these issues, we introduce **VideoVerse**, a comprehensive benchmark that focuses on evaluating whether a T2V model could understand complex temporal causality and world knowledge in the real world. We collect representative videos across diverse domains (e.g., natural landscapes, sports, indoor scenes, science fiction, chemical and physical experiments) and extract their event-level descriptions with inherent temporal causality, which are then rewritten into text-to-video prompts by independent annotators. For each prompt, we design a suite of binary evaluation questions from the perspective of dynamic and static properties, with a total of ten carefully defined evaluation dimensions. In total, our VideoVerse comprises 300 carefully curated prompts, involving 815 events and 825 binary evaluation questions. Consequently, a human preference aligned QA-based evaluation pipeline is developed by using modern vision-language models. Finally, we perform a systematic evaluation of state-of-the-art open-source and closed-source T2V models on VideoVerse, providing in-depth analysis on how far the current T2V generators are from world models.

## 1 Introduction

Text-to-video (T2V) models can translate natural language into coherent and high-quality content videos, unlocking new possibilities for human–AI interaction and multimodal media creation, which have a wide range of applications, including creative content generation (Kong et al., 2024; Wan et al., 2025; Yang et al., 2024), virtual reality (Akimoto et al., 2022), and video editing (Geyer et al., 2024; Yoon et al., 2024), etc. Along with the growing impact of T2V models, how to rigorously evaluate them has become critically important for benchmarking progress and guiding model construction, training, and deployment. Early T2V benchmarks, such as VBench (Huang et al., 2024a) and EvalCrafter (Liu et al., 2024), primarily evaluate the generated videos at the frame level, focusing on aesthetic quality and image fidelity. Subsequent benchmarks focus on assessing semantic alignment, i.e., whether the generated video content matches the given text prompt. Recent benchmarks such as VBench2 (Zheng et al., 2025) extend the evaluation to intrinsically complex semantic alignment through VLM-based question answering, while Video-Bench (Han et al., 2025) performs the evaluation by comparing detailed video-to-text captions with the original instructions.

However, the rapid advancement of T2V technologies (Kong et al., 2024; Wan et al., 2025; Yang et al., 2024) has begun to expose the limitations of existing benchmarks. State-of-the-art T2V models have not only demonstrated strong instruction-following abilities (Ma et al., 2025; Chen et al., 2025a), but also exhibited the capacity to understand world knowledge, such as temporal and causal relations among events (Google DeepMind, 2025), while producing cinematic quality videos (Xiao et al., 2025; Gao et al., 2025). As a consequence, existing T2V benchmarks are becoming insuf-

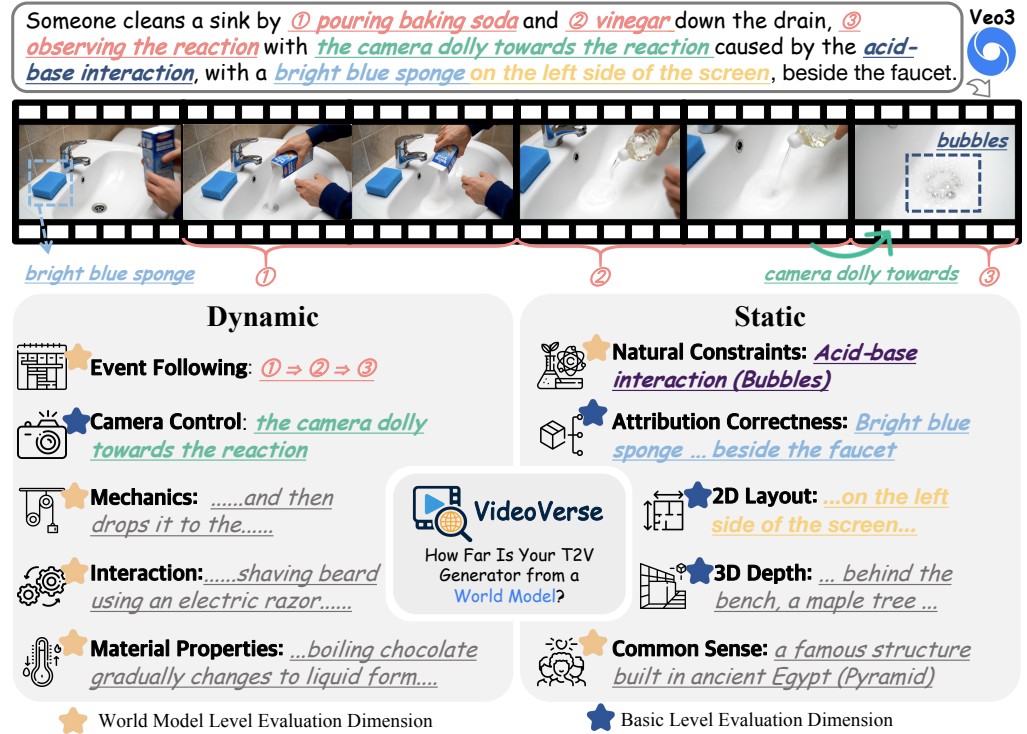

Figure 1: Overview of the evaluation dimensions of **VideoVerse**, which are considered from two complementary perspectives: the **Dynamic** and the **Static**. Under the two categories, a total of ten dimensions, which include six world model level evaluation dimensions and four basic level evaluation dimensions.

ficient to measure the emerging capabilities of modern T2V models and differentiate their performance for two reasons. First, their prompts can only specify the expected content, lacking the ability to infer or predict events not explicitly described. Second, complex world knowledge and common sense reasoning remain largely unexplored in existing benchmarks. Considering that T2V models are becoming the key elements in building "world models", which could not only generate visual content but also simulate, reason and predict the dynamics of the physical world (Agarwal et al., 2025; Kang et al., 2024), there is a strong demand to develop a new benchmark, which can evaluate the capabilities of T2V generators in terms of a "world model".

To address the challenges mentioned above, we propose **VideoVerse**, a comprehensive benchmark designed to assess modern and emerging T2V models. As illustrated in Fig. 1, VideoVerse contains a set of carefully designed evaluation dimensions from two complementary perspectives: **dynamic** (which should be presented across temporal frames) and **static** (which could be presented in a single frame). In order to effectively evaluate the capabilities of T2V models in terms of a world model, we consider two crucial aspects. The first is temporal causality at the event level, which measures whether a T2V model can produce a series of events with strong causal relationships. A set of *Event Following* prompts are designed to assess the T2V model along this dimension. The second aspect includes a series of dimensions to evaluate whether a T2V model can understand the natural world. From the static perspective, we introduce *Natural Constraints* and *Common Sense*, which play important roles in our lives. Here, *Common Sense* refers to socially shared conventions (e.g., "the representative animal of Sichuan Province, China is the panda" (Wikipedia, 2025a)), while *Natural Constraints* captures physical or chemical laws of the natural world (e.g., "concentrated sulfuric acid carbonises wood upon contact" (Wikipedia, 2025c)). In the dynamic perspective, in addition to *Event Following*, we consider *Mechanics*, *Interaction*, and *Material Properties*, which are common dynamic properties of the real world. Furthermore, as a T2V benchmark, it should also consider the basic abilities required for T2V models, including *Camera Control* in the dynamic perspective, and *Attribution Correctness*, *2D Layout*, and *3D Depth* in the static perspective. Finally, a total of ten dimensions (five dynamic and five static) are defined in our VideoVerse benchmark.

To better align with the world model's ability, we design all the prompts with the guideline of "hidden semantics" (i.e., the model should predict and generate the expected scene beyond the given prompts), which requires the model to understand the physics and natural laws of the real world. For example, with the prompt shown in Fig. 1, the model should generate content that produces bubbles when baking soda and vinegar react in the sink. In conjunction with our prompt design, we further propose a QA-based evaluation pipeline, which simulates human-like evaluation process based on powerful VLMs (Wei et al., 2025; Han et al., 2025).

Overall, with 300 carefully curated high-quality prompts, VideoVerse contains 815 events with 825 binary evaluation questions that cover different dimensions. Unlike previous benchmarks, where each prompt corresponds to a single evaluation dimension, the prompts in VideoVerse integrate multiple evaluation aspects in one prompt. This design not only provides richer and more challenging prompts for T2V models but also enables a more cost-effective evaluation procedure.

The main contributions of our work are summarized as follows. First, we present **VideoVerse**, a high-quality, carefully curated benchmark with 300 prompts covering ten dimensions that span from basic instruction following to world model level reasoning. Second, we conduct an extensive evaluation of both open-source and closed-source T2V models, showing that while they perform comparably on traditional benchmarks, their performance differs largely on VideoVerse, especially in dimensions requiring causal reasoning and world knowledge. Third, our analysis demonstrates that current T2V models still fall short of world models, indicating new challenges and directions for future research in this rapidly evolving field.

## 2 RELATED WORK

**Text-to-Video (T2V) Models.** Earlier T2V models (Ho et al., 2022; Hong et al., 2023; Chen et al., 2023; Wang et al., 2024) are limited to short clips with limited expressiveness, while recent models have demonstrated substantial improvements by leveraging larger backbones and higher quality training data (Kong et al., 2024; Wan et al., 2025; Ma et al., 2025; Chen et al., 2025a; Esser et al., 2023; Zheng et al., 2024; Peng et al., 2025). HunyuanVideo (Kong et al., 2024) and WanX (Wan et al., 2025) employ DiT-based architectures and considerably enhance the performance of open-source models, while StepVideo (Ma et al., 2025), with its 30B parameters, achieves state-of-the-art results across multiple dimensions. Meanwhile, closed-source models generally outperform their open-source counterparts (Google DeepMind, 2025; Brooks et al., 2024; Pika Lab, 2025; Kling, 2025; MiniMax, 2025), particularly in video length, visual fidelity, and adherence to textual instructions. The rapid progress of T2V models highlights their potentials to become world models, which, however, poses challenges on how to accurately evaluate their performance along such perspectives.

**T2V Model Evaluation.** Earlier T2V benchmarks (Huang et al., 2024a; Liu et al., 2024; Huang et al., 2024b) rely primarily on frame-level aesthetic and image quality metrics such as FID (Seitzer, 2020), FVD (Unterthiner et al., 2019), IS (Salimans et al., 2016), and basic video attributes such as subject consistency. With the rapid development of T2V models, frame quality has reached human-perceptual standards. Benchmarks shifted their focus to assessing whether the generated video content matches the given text prompt. For example, VBench2 (Zheng et al., 2025) employs VLM-based question answering and expert models to evaluate complex semantic alignment, while Video-Bench (Han et al., 2025) shifts its evaluation entirely into the textual space by aligning video-to-text captions with instructions. StoryEval (Wang et al., 2025a), an event-centric T2V benchmark, evaluates whether the events described in the prompt will occur, but it neglects the temporal causality among events and the static attributes of videos. Howerver, the prompts in these benchmarks lack a "hidden semantics" guideline (i.e., the model should generate expected behaviours beyond what is explicitly stated) and fail to systematically incorporate world knowledge (Niu et al., 2025).

**Evaluation of T2V Models as "World Model".** T2V models have shown the potential to be "world models" (Huang et al., 2025; Chen et al., 2025b), yet they still struggle to generate realistic content aligned with the real world (Wang et al., 2025b; Kang et al., 2024). Some works (Bansal et al., 2024; 2025; Li et al., 2024; Meng et al., 2024) focus on evaluating T2V models' ability to capture physical laws or simulate the real world. However, these studies consider only physical regularities, particularly motion laws, overlooking broader aspects of world knowledge, such as state changes, chemical interaction, cultural common sense, etc. In addition, previous T2V Benchmarks (Zheng et al., 2025; Huang et al., 2024a; Liu et al., 2024) only consider the content explicitly described in

the prompt, but lack the ability to evaluate the T2V models in terms of hidden semantics, which are primitive elements of a "World Model".

To this end, we introduce VideoVerse, a comprehensive benchmark designed for evaluating modern T2V models' ability from basic instruction following to world model level understanding with a simple, easy-to-use, and human-preference-aligned evaluation protocol. In addition, our compact prompt design enables efficient assessment with a limited number of well-designed prompts, a property that is particularly useful for benchmarking increasingly large T2V models.

## 3 CONSTRUCTION OF VIDEOVERSE BENCH

Unlike existing T2V benchmarks, which construct prompts in a relatively straightforward manner to cover a wide range of visual elements, we argue that a world model level T2V models should not only generate videos that align with the text prompt (e.g., whether the objects appear or whether the attributes such as color and texture are correct), but also demonstrate strong capabilities in understanding the implicit temporal and logical relations among events, as well as the world knowledge such as *Natural Constraints* and *Common Sense*. To this end, we define ten evaluation dimensions to assess the quality of generated videos from both static and dynamic perspectives, as illustrated in Fig. 1 and detailed in the following sections.

### 3.1 STATIC DIMENSIONS

Static dimensions refer to those facts or properties that can be evaluated from a single frame. The following five static dimensions are defined in VideoVerse:

**Natural Constraints**, which evaluates whether the generated content adheres to natural scientific laws. For example, a lake at $-20°$C should be frozen.

**Common Sense**, which evaluates whether the generated content aligns with the cultural or contextual common sense knowledge implied in the prompt. For example, a "tree representative of Japanese culture" should be a cherry blossom tree.

**Attribution Correctness**, which evaluates whether the objects mentioned in the prompt appear in the generated video, and whether their specified attributes, such as colour, material and shape, are correctly generated.

**2D Layout**, which evaluates whether the 2D spatial arrangement among the objects mentioned in the prompt is correctly represented.

**3D Depth**, which evaluates whether the perspective relationships among the objects mentioned in the prompt, such as which objects are in the foreground or background, are correctly represented.

### 3.2 DYNAMIC DIMENSIONS

Dynamic dimensions refer to those facts or properties that can only be evaluated by understanding the temporal dynamics in a video (which cannot be represented in a single frame). The following five dynamic dimensions are defined in VideoVerse:

**Event Following**, which evaluates whether the T2V model generate a temporally causal sequence of events specified in the prompt.

**Mechanics**, which evaluate whether an object in the video follows the laws of mechanics in its motion. This does not require interaction with other objects. For example, when a dumbbell is released from the hand, it should fall to the ground due to gravity, without a change in shape.

**Interaction**, which evaluates whether the interactions between objects are physically reasonable. Here, our focus is on interactions that involve direct object contact, regardless of material properties. For example, shaving with a razor should result in shorter facial hair.

**Material Properties**, which evaluate whether an object's behaviour is consistent with its intrinsic material properties, even without explicit contact with other objects. For example, chocolate should gradually melt when heated.

**Camera Control**, which evaluates whether the camera operations specified in the prompt, such as focus control and motion trajectory, are executed correctly.

## 3.3 PROMPT CONSTRUCTION

The prompts in our VideoVerse are drawn from three distinct domains with different objectives: (1) Daily Life, (2) Scientific Experiment, and (3) Science Fiction. Each domain undergoes a tailored preprocessing pipeline, as detailed below.

**Daily Life:** We sample a large set of videos from the real-world dataset ActivityNet Caption (Krishna et al., 2017). Although ActivityNet Caption provides captions with multiple events, which is consistent with the event-centric design philosophy of our VideoVerse, it suffers from two limitations: (i) not all events exhibit temporal causality, which is a crucial requirement in our benchmark, and (ii) its captions often correspond to overly long video segments, making some of them unsuitable for constructing concise T2V prompts. To address these issues, we use GPT-4o to filter and refine the original captions, yielding suitable prompts for this domain.

**Scientific Experiment:** Although other categories of prompts occasionally include prompts related to natural science, they are not explicitly designed for this purpose. To better evaluate the world model capabilities of T2V models, we manually collect a set of prompts derived from high-school-level natural science experiments from the web and incorporate them into this domain.

**Science Fiction:** Unlike the first two domains, which focus on realistic scenarios, this category focuses on imaginative, non-realistic content. It is designed to test T2V models' out-of-domain generalization, as training corpora rarely contain fictional scenarios. We curate science-fiction prompts from VidProM (Wang & Yang, 2024), a community-collected dataset, and apply GPT-4o to clean irrelevant tokens. The resulting set forms the source prompts for this domain.

After collecting source prompts from the three domains, we employ a unified pipeline that takes advantage of GPT-4o as an event-causality extractor, which is illustrated in Fig. 4 in **Appendix A**. Specifically, GPT-4o identifies causal relationships between events within a video and organizes them into an initial *raw prompt*. However, these raw prompts only capture event-level structures, but do not include the full range of dimensions required for evaluation. To mitigate these issues, we invite independent human annotators to enrich the raw prompts with appropriate semantic content for the relevant evaluation dimensions, refining them into the final T2V prompts. This manual process ensures prompt quality and fairness of the evaluation. Specifically, for each raw prompt (comprising events extracted by GPT-4o), independent annotators select the most appropriate evaluation dimensions and revise the raw prompt accordingly. Each added dimension is paired with a corresponding binary evaluation question. Consider that VideoVerse requires annotations to capture hidden semantics and world-model-level knowledge, all annotators hold at least a bachelor's degree. Furthermore, to balance different evaluation dimensions, annotators periodically review their prior annotations and adjust subsequent labeling preferences to reduce bias. Beyond ensuring fairness, manual refinement also provides interpretability and reliability that cannot be guaranteed by fully automated methods, strengthening the credibility of our VideoVerse.

## 3.4 EVALUATION PROTOCOL

Traditional benchmarks primarily rely on various expert models to evaluate frame-level image quality. Subsequent benchmarks employ large vision-language models (VLMs) for QA-based evaluation; however, these VLMs have limited video understanding capabilities. Moreover, in these benchmarks, the text instruction of the generated video is directly exposed to the VLM, which often leads to hallucination issues, i.e., the VLM assumes that certain elements mentioned in the instruction will appear in the video even if they do not.

Unlike previous benchmarks, VideoVerse not only assesses the fundamental capabilities of T2V models but also measures their gap to a comprehensive world model. To enable such a complex and holistic evaluation, we leverage state-of-the-art VLMs with rich world knowledge and strong reasoning abilities. Different from previous works, which expose the full text instruction to the VLM, we provide dimension-specific binary questions to the VLM, thereby mitigating the hallucination issues. In particular, our VLM-based evaluation protocol consists of two components.

**Temporal Causality Evaluation**. *StoryEval* (Wang et al., 2025a) also evaluates T2V models based on event-level reasoning. However, it only verifies whether the described events are generated, without considering the temporal and causal correlations among them.

We adopt the **L**ongest **C**ommon **S**ubsequence (**LCS**) algorithm (Wikipedia, 2025b) as the evaluation protocol for *Event Following* performance. Specifically, we first use a powerful VLM to identify whether each event occurs in the generated video and extract the corresponding sequence of events. Let the ground truth event sequence be $E = \{e_1, e_2, \ldots, e_n\}$ and the predicted event sequence be $\hat{E} = \{\hat{e}_1, \hat{e}_2, \ldots, \hat{e}_{\hat{m}}\}$, where events absent from the generated video are not output by the VLM. We then compute the longest subsequence of $\hat{E}$ that aligns with $E$, and take its length as the *Event Following* score of the generated video.

**Prompt-specific Dimension Evaluation**. As shown in Fig. 1, VideoVerse defines a total of ten evaluation dimensions. Apart from the *Event Following* dimension, which is included for every prompt, the other dimensions vary across prompts, and the number of binary questions associated with each dimension is also varying. For the $m$ binary evaluation questions beyond *Event Following*, we conduct $m$ independent interactions with the VLM, and the number of correctly answered questions is taken as the score for these additional dimensions.

Given the characteristics of our evaluation protocol and following previous works (Fu et al., 2024), we adopt **a cumulative scoring strategy** rather than a percentage score as the final evaluation metric for a T2V model in VideoVerse. Suppose that a prompt $P$ contains $N$ evaluation dimensions (excluding *Event Following*), where the $i$-th dimension is associated with $k_i$ binary evaluation questions. Let model $M$ generate a video $V$ under prompt $P$. The score of $V$ is defined as:

$$S(V) = \text{LCS}(V) \ + \ \sum_{i=1}^{N} \sum_{j=1}^{k_i} \mathbb{I}\big(\text{Eval}(V, q_{i,j}) = \text{Yes}\big), \tag{1}$$

where $\text{LCS}(V)$ denotes the LCS score for *Event Following*, $q_{i,j}$ is the $j$-th binary evaluation question under the $i$-th dimension, and $\mathbb{I}(\cdot)$ is the indicator function that equals 1 if the condition holds and 0 otherwise. Thus, the final score of model $M$ on prompt $P$ is given by $S(V)$. We provide the prompts used in the evaluation in the **Appendix B**.

### 3.5 Comparison with other T2V Benchmarks

Early T2V benchmarks primarily evaluate video quality using domain-specific expert models with frame-level aesthetic and image quality metrics. Later benchmarks shift their focus toward assessing whether the generated video content matches the given text prompt. However, these benchmarks lack the ability to evaluate T2V generators from the perspective of a world model. Compared with previous benchmarks, VideoVerse substantially increases the complexity of prompts by introducing highly diverse scenes, characters, and event content. Each prompt contains events with implicit causal and temporal relations, while incorporating rich world knowledge.

To quantify the distinctions introduced by the VideoVerse prompt set, we use CLIP to extract semantic embeddings of prompts from mainstream benchmarks and compute their cosine similarity, as shown in Fig. 2. The results reveal that existing benchmarks contain a considerable degree of semantic redundancy, whereas VideoVerse achieves the highest semantic uniqueness. Moreover, the prompts in existing benchmarks are overly simplistic in terms of length and cannot represent the complex instructions that users typically provide to world model level T2V systems. In contrast, thanks to its careful and diverse design, VideoVerse not only exhibits a significantly longer average prompt length than existing benchmarks but also demonstrates a more natural length distribution.

## 4 Experiments

In this section, we evaluate the major T2V models on **VideoVerse** and provide detailed analyses of their performance. We also conduct a user study to examine whether our evaluation protocol is sufficient and aligned with human perception. The T2V models we evaluated include CogVideoX1.5-5B (Yang et al., 2024), SkyReels-V2-14B Chen et al. (2025a), HunyuanVideo (Kong et al., 2024), OpenSora2.0 (Peng et al., 2025), WanX2.1-14B (Wan et al., 2025), WanX2.2-14B (Wan et al., 2025), Hailuo (MiniMax, 2025) and Veo3 (Google DeepMind, 2025). The deployment details of those models can be found in **Appendix C**.

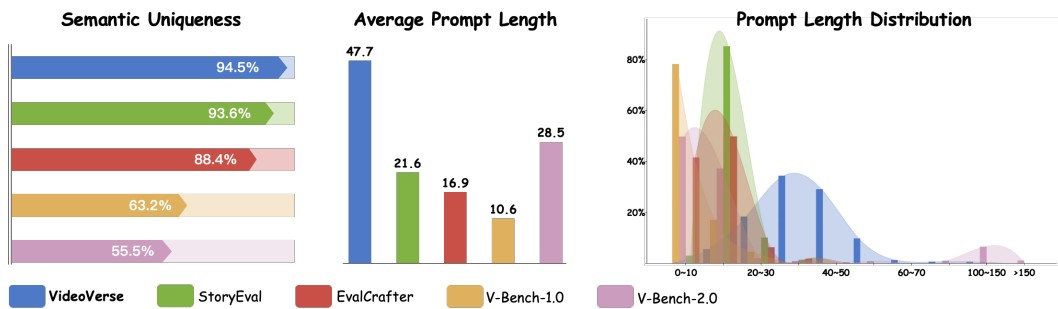

Figure 2: **Left**: We extract CLIP embeddings of prompts from mainstream T2V benchmarks and compute their cosine similarity. We see that existing benchmarks contain a large number of redundant prompts. **Middle**: Users typically provide complex instructions when interacting with world model level T2V systems, yet existing benchmarks generally consist of overly short prompts. **Right**: The prompt length distribution of VideoVerse aligns closely with natural usage patterns. We also provide more comparisons with other T2V benchmarks in the **Appendix D**.

Table 1: Performance of Open-Source and Closed-Source models on VideoVerse based on Gemini2.5 pro, with the best values highlighted in bold. The light green columns represent the world model level dimensions, while the other columns represent the basic level dimensions. We employ the 5B variant of CogVideoX1.5 and 14B variant of SkyReels-V2. (S) and (L) denote the "Short Video" (5s) and "Long Video" (10s) settings, respectively.

| Model | Overall | Dynamic | | | | | Static | | | | |
|---|---|---|---|---|---|---|---|---|---|---|---|
| | | Event Following | Camera Control | Interaction | Mechanics | Material Properties | Natural Constra. | Common Sense | Attr. Correct. | 2D Layout | 3D Depth |
| *Open-Source Models* | | | | | | | | | | | |
| CogVideoX1.5 (S) | 922 | 424 | 37 | 37 | 25 | 26 | 36 | 41 | 178 | 66 | 52 |
| CogVideoX1.5 (L) | 916 | 426 | 38 | 38 | 28 | 22 | 38 | 38 | 183 | 58 | 47 |
| SkyReels-V2 (S) | 963 | 484 | 43 | 37 | 30 | 22 | 32 | 43 | 161 | 61 | 50 |
| SkyReels-V2 (L) | 997 | 511 | 37 | **42** | 33 | 24 | 36 | 36 | 169 | 62 | 47 |
| Wan2.1-14B | 998 | 496 | 43 | 34 | 32 | 24 | 35 | 46 | 168 | **68** | **52** |
| Hunyuan | 923 | 446 | 39 | 32 | 34 | 25 | 37 | 42 | 160 | 60 | 48 |
| OpenSora2.0 | 1015 | 482 | 48 | 36 | 29 | 27 | **48** | **50** | 182 | 62 | 51 |
| Wan2.2-A14B | **1112** | **567** | **61** | 36 | **39** | **30** | 37 | 44 | **185** | 64 | 49 |
| *Closed-Source Models* | | | | | | | | | | | |
| Minimax-Hailuo | 1241 | 623 | 76 | 44 | 42 | 36 | 55 | 53 | **188** | **69** | **55** |
| Veo-3 | **1334** | **680** | **77** | **54** | **50** | **36** | **68** | **58** | 188 | 68 | **55** |

## 4.1 Main Results

Tab. 1 presents the evaluation results of the T2V models on VideoVerse using Gemini 2.5 Pro (Comanici et al., 2025). Among open-source models, while Wan2.2-14B achieves the highest overall score, the best performers across **world model** level dimensions diverge. OpenSora2.0 demonstrates strong results in *Common Sense* and *Natural Constraints* in the static category. This can be attributed to its design: unlike other T2V models, OpenSora2.0 conditions video generation on the outputs from the powerful T2I model Flux (Labs et al., 2025), which significantly enhances its generation capability along static world model level dimensions. In contrast, SkyReels-V2 (L) achieves the best performance in *Interaction*, which is because, compared with other dynamic dimensions, *Interaction* emphasises the interactions between objects; longer generation length provides it with a broader context to model such behaviours. For the remaining world model level dimensions, WanX2.2-A14B outperforms the other open-source models for its advanced architecture and large-scale training data. Across the **basic** level dimensions, most open-source models perform comparably except for the *Camera Control* dimension, which requires strong instruction-following capability.

However, open-source models still lag considerably behind closed-source systems in all evaluation dimensions. Veo-3 achieves the best overall performance, establishing state-of-the-art results in most dimensions. Similar to open-source systems, closed-source models show comparable performance on the basic level dimensions. However, their abilities diverge markedly at the world model level, reflecting the fact that even for the most advanced closed-source models, reasoning about the world remains a substantial challenge. We will discuss this further in Sec. 4.2.

Table 2: Performance gap between open-source and closed-source T2V models. Open-source models exhibit comparable performance to closed-source models on basic dimensions, whereas the gap is more pronounced on world-model dimensions. Since the *Event Following* (EF) dimension uses LCS as its metric score, we also present the statistics without EF (w/o EF). Notably, even the advanced closed-source model Veo-3 has much room to improve to be a world model.

| Model | Basic | World Model w/o EF | World Model w EF | Overall Score |
|---|---|---|---|---|
| *Open-Source* **Models** | | | | |
| CogVideoX1.5(L) | 326 △ -62 | 164 △ -102 | 590 △ -356 | 916 △ -418 |
| Wan2.2-A14B | 359 △ -29 | 186 △ -80 | 753 △ -193 | 1112 △ -222 |
| *Closed-Source* **Models** | | | | |
| Minimax–Hailuo | 388 △ 0 | 230 △ -36 | 853 △ -93 | 1241 △ -93 |
| **Veo-3** | **388 (out of 477)** | **266 (out of 348)** | **946 (out of 1163)** | **1334 (out of 1640)** |

## 4.2 DISCUSSIONS

**Will Video Length Influence Event Following Performance?** All prompts in our VideoVerse contain at least one event that requires T2V models to generate content along the temporal dimension. Although modern T2V models can generate longer videos than earlier ones, the length is still limited to a few seconds. This raises a question: Is their limited performance on *Event Following* primarily due to the short video length, which restricts the number of events that can be generated?

Based on our experimental results, the answer is **No**. As shown in Tab. 1, Veo-3 produces only 8-second videos, yet it consistently outperforms models with 10-second outputs across all dimensions. Moreover, for models capable of generating 10-second videos, their performance on *Event Following* shows no clear advantage over shorter ones (e.g., CogVideoX1.5-S/L, SkyReels-V2-S/L). Notably, the open-source Wan2.2-A14B, limited to 5-second outputs, still exceeds other open-source models in *Event Following*, including those generating 10-second videos. Therefore, for the prompts in VideoVerse, the current length of generated videos already provides sufficient temporal capacity for T2V models to process the required events.

**How Far are the Current T2V Models from a "World Model"?** We show the performance of typical open-source and closed-source T2V models in terms of basic abilities and world model level abilities in Tab. 2. We see that they perform comparably in basic ability, but their gap in world model level abilities is much larger. The powerful Veo3 model demonstrates the best performance in world knowledge. It achieves 266 points (w/o EF) out of the total of 348 world-model scores (76.4%, 266/348), while it achieves 388 points out of the total of 477 scores (81.3%, 388/477) in basic dimensions. The statistics for each category of evaluation dimensions are provided in **Appendix D**). This observation indicates that, despite the impressive generative capabilities of current T2V models, they are still far from achieving the level of a true "World Model."

There are two main limitations of current T2V models. i) **"Hidden" Semantics Following.** T2V models often restrict their generation to the surface-level semantics explicitly mentioned in the prompt, while ignoring implicit or hidden semantics beyond the text. ii) **Limited Understanding** of the real world. Although current models can sometimes generate contents explicitly presented in prompts, they often fail to generate reasonable output when additional semantic constraints grounded in real-world knowledge are introduced. Please refer to **Appendix E** for some cases.

## 4.3 CASE STUDY AND USER STUDY

Fig. 3 presents a case from VideoVerse. We see that the closed-source model Veo-3 not only successfully generates all events but also correctly understands common sense knowledge: "the steed of Tang Sanzang......" refers to a white horse. In contrast, open-source models such as Wan 2.1 and Hunyuan generate content accurately in the *Attribution Correctness* dimension but struggle with world model level dimensions, such as *Event Following* and *Com-*

Table 3: User study results: Gemini 2.5 Pro demonstrates high consistency with human judgment in evaluating T2V models on VideoVerse.

| Evaluation Type | Question Number | Consistence Ratio |
|---|---|---|
| Basic Binary Question | 231 | 90.47 |
| World Model Binary Question | 198 | 94.37 |
| Event Following | 165 | 94.26 |

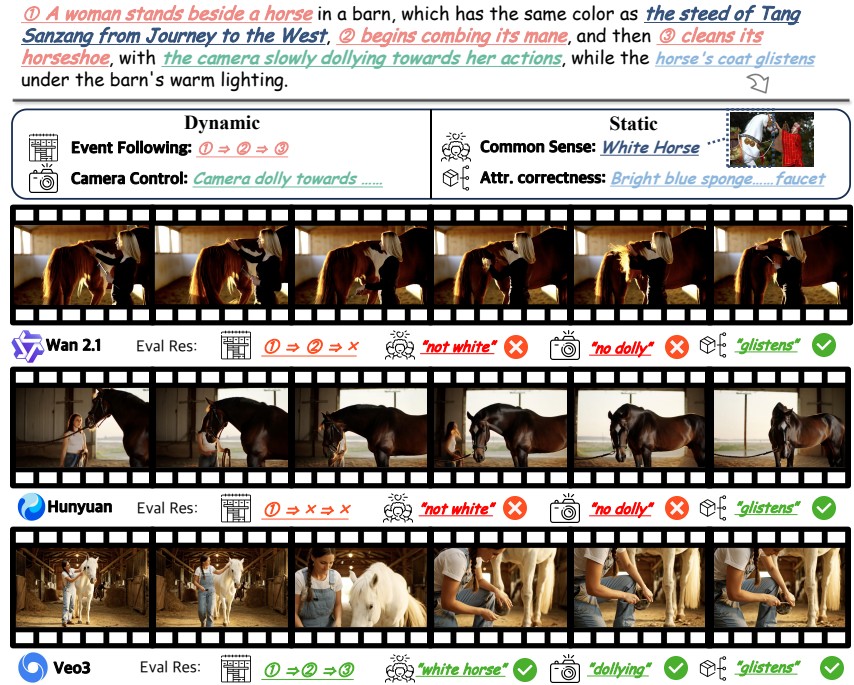

Figure 3: Case study of T2V models' performance on our VideoVerse. Gemini 2.5 Pro is used as the evaluator. Wan 2.1 and Hunyuan successfully generate the corresponding attribution content (horse's coat glistens) but struggle with *Event Following* and *Common Sense*, whereas Veo-3 demonstrates strong performance across all dimensions.

*mon Sense*. This highlights the significant gap between open-source and closed-source models at the world model level dimensions.

We employ the SOTA video understanding VLM, Gemini 2.5 Pro, to evaluate the T2V models on VideoVerse. To examine how well Gemini 2.5 Pro aligns with human judgment, we conduct a user study using 15 videos generated from 11 prompts in our VideoVerse, spanning over the ten evaluation dimensions. A total of 11 volunteers are invited to participate in the study, and there are 594 questions in the evaluation. Following the same protocol as in Sec. 3.4, participants are provided with the video and the corresponding question, mirroring the VLM evaluation setting. As shown in Tab. 3, there is a high consistency (>90%) between human judgment and VLM evaluation in different dimensions. More details are provided in **Appendix F**. Although Gemini2.5 Pro demonstrates high alignment with human judgment, it is a closed-source commercial model that has limited accessibility to the community. To facilitate broader adoption of VideoVerse, we also provide results using other open-source VLMs as evaluators, as detailed in **Appendix G**.

## 5 CONCLUSION

In this paper, we presented **VideoVerse**, the first benchmark designed to evaluate T2V models from a world model perspective, which has 300 carefully constructed prompts from 10 well-designed evaluation dimensions. Through systematic evaluation, we find that although substantial progress has been made by current T2V models, they still have gaps in terms of world models. Meanwhile, the performance of open-source T2V models is much lower than that of their closed-source counterparts. Our findings highlighted the challenges of T2V research and underscored the limitations of current T2V models in understanding, reasoning, and generalising in dynamic environments. We hope that VideoVerse can facilitate the development of next-generation T2V models.

**Limitations**. While VideoVerse is the first attempt to evaluate T2V models through the lens of a world model, it has several limitations. First, the definition of a "World Model" is inherently multi-faceted, and our benchmark captures only a subset of key dimensions. Moreover, VideoVerse currently focuses on 2D T2V models, whereas emerging research on 3D and open-world generation highlights other critical aspects of world modeling.

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

# Appendix

## Table of Contents

# A  THE PIPELINE OF PROMPT CONSTRUCTION

As illustrated in Fig. 4, the construction pipeline of our prompts begins with three source domains: VidProM (science fiction), ActivityNet (daily life), and web-collected high school level experiments. Each domain undergoes domain-specific processing to generate raw-prompt pools. Subsequently, GPT-4o is employed to rewrite these prompts and extract temporally related events. Finally, independent annotators refine the outputs by incorporating one or more evaluation dimensions while preserving the original event structure, resulting in the final prompts used in VideoVerse.

# B  EVALUATION PROMPTS

## B.1  BINARY EVALUATION QUESTION

In VideoVerse, all evaluation dimensions, except *Event Following*, are assessed using binary questions. The prompts for these questions are listed in Tab. 4. After obtaining the VLM's response, we extract the final answer ("Yes" or "No") by regular expressions.

## B.2  EVENT EVALUATION QUESTION

For the *Event Following* dimension, the corresponding evaluation prompt is shown in Tab. 5. To ensure robust parsing and avoid ambiguity in free-form responses, we instruct the VLM to enclose its output within ⟨output⟩ and ⟨/output⟩ tags. The enclosed content is then extracted using regular expressions for evaluation.

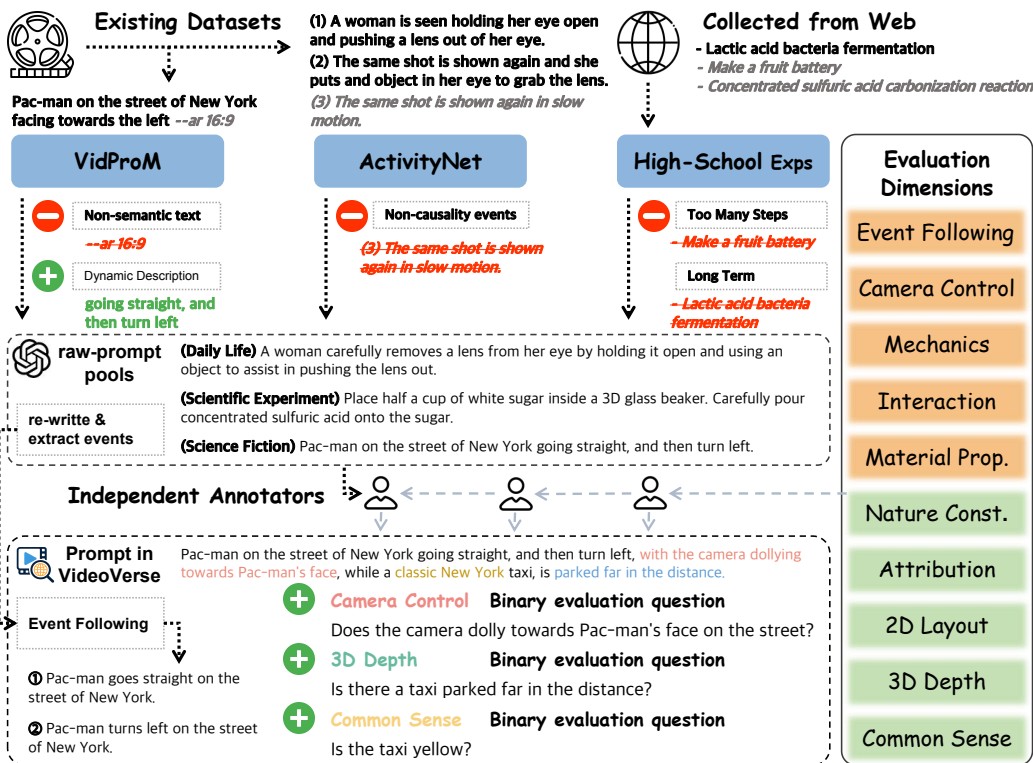

Figure 4: Prompt construction pipeline of VideoVerse. Source prompts are drawn from three domains: science fiction (VidProM), daily life (ActivityNet), and human collected high school level experiments. After domain specific filtering, GPT-4o extracts temporally related events to form raw prompts. Independent annotators then refine these raw prompts by incorporating one or more evaluation dimensions, while preserving the original event structure, to produce the final prompts.

Table 4: Prompt for Binary Question Evaluation used in our VideoVerse.

> **Prompt for Binary Question Evaluation**
>
> "In this video."
> "{BINARY QUESTION}"
> "Answer with Yes or No."

Table 5: Prompt for *Event Following* evaluation used in our VideoVerse. The VLMs (e.g., Gemini2.5 Pro) respond with the existing events' index in order, which is then used to calculate the LCS with the ground truth event order.

> **Prompt for Event Following Evaluation**
>
> "These are several events that may happened in the video."
> "Please sort them in the order in which they occurred. "
> "If an event does not occur, the index corresponding to the event is ignored."
> "Only output the corresponding letters in order, use commas to separate, and wrap your response with ⟨output⟩⟨/output⟩. Such as ⟨output⟩B,D⟨/output⟩ or ⟨output⟩A,B,C⟨/output⟩"
> "{*EVENTS*}"

## C    EVALUATION DETAILS

### C.1    OPEN-SOURCE T2V MODELS

For the open-source T2V models evaluated in this paper, all experiments are conducted on servers with NVIDIA A800 GPUs. The sources of the T2V models are as follows:

- **CogVideoX1.5**: accessed from CogVideoX Official GitHub.

- **OpenSora2.0**: accessed from OpenSora Official GitHub.

- **SkyReels-V2**: accessed from SkyReels-V2 Official GitHub.

- **Wan2.1-14B**: accessed from Wan2.1-14B Official GitHub, with checkpoints and inference code from Hugging Face.

- **Wan2.2-A14B**: accessed from Wan2.2-A14B, with checkpoints and inference code from Hugging Face.

- **Hunyuan**: accessed from Hunyuan Official GitHub.

Furthermore, as discussed in Sec. 1 of the main paper, the rapid development of T2V models has led to a significant increase in inference time. We report the video inference time along with the GPUs used for each model in Tab. 6. We can see that it will take 648,000 seconds (about one week) to finish the inference of Skyreels-V2(L) on VideoVerse (300 prompts). This also motivates the design of our VideoVerse, where multiple evaluation dimensions are incorporated for each prompt to better capture the trade-offs in efficiency and evaluation.

### C.2    CLOSED-SOURCE T2V MODELS

For closed-source models, we access them exclusively through their official APIs: Minimax-Hailuo via [1] and Veo-3 via [2]. Due to the high cost of Veo-3, we employ Veo-3-fast in our experiments.

---

[1] https://www.minimax.io/platform/document/video_generation
[2] https://developers.googleblog.com/en/veo-3-now-available-gemini-api/

Table 6: Single video inference time and the used GPUs of open-source T2V models.

| Model | Inference Time (s) | Number of GPUs (A800) |
|---|---|---|
| CogVideoX1.5(S) | ∼415 | 1 |
| CogVideoX1.5(L) | ∼869 | 1 |
| SkyReels-V2(S) | ∼720 | 4 |
| SkyReels-V2(L) | ∼2160 | 4 |
| Wan2.1 | ∼948 | 1 |
| Hunyuan | ∼1102 | 1 |

Table 7: Statistics of our VideoVerse. The light gray rows represent the **world model** level dimensions, while the other rows represent the **basic** level dimensions.

| Evaluation Category | Event Number |
|---|---|
| Event Following | 815 |

| Evaluation Category | Binary Question Number |
|---|---|
| Natural Constraints | 85 |
| Common Sense | 77 |
| Interaction | 69 |
| Mechanics | 68 |
| Material Properties | 49 |
| Attribute Correctness | 217 |
| Camera Control | 116 |
| 2D Layout | 86 |
| 3D Depth | 58 |

| Overall Evaluation Number | Number |
|---|---|
| World Model Level Evaluation w/o EF | 348 |
| World Model Level Evaluation w/ EF | 648 |
| Basic Level Evaluation | 477 |

| Evaluation Density | Number |
|---|---|
| Avg. Dimensions / Prompt | 3.75 |

# D    STATISTICS OF VIDEOVERSE

## D.1    STATISTICS OF EACH EVALUATION DIMENSION

Tab. 7 summarizes the detailed statistics of VideoVerse. The VideoVerse includes six world model level evaluation dimensions: *Material Properties*, *Natural Constraints*, *Common Sense*, *Mechanics*, *Interaction*, and *Event Following*, which are designed to assess T2V models from a world model perspective. In addition, VideoVerse incorporates four basic level dimensions: *Attribute Correctness*, *2D Layout*, *3D Depth*, and *Camera Control*, which aim to evaluate the fundamental abilities of a T2V model. Among them, *Camera Control* is particularly challenging as it requires strong instruction-following capability.

## D.2    THE DESIGN OF "HIDDEN SEMANTICS"

As emphasized in Sec. 1 of the main paper, a key design of our VideoVerse is the "hidden semantics" within the prompts. To illustrate this, we compare VideoVerse with the most recent T2V benchmark, VBench2.0. As shown in Tab. 8, VBench2.0 often incorporates explicit descriptions of physical phenomena in the prompt itself (e.g., specifying that a water droplet remains "spherical" due to surface tension). However, such details should not be explicitly provided in the prompt, as they should be inferred by the T2V model as world knowledge. In contrast, VideoVerse intentionally hides these semantics within the prompt, thereby requiring models to infer and generate them based on their learned world modeling ability, rather than textual guidance.

Table 8: Comparison of prompts between VBench2.0 and our VideoVerse. Unlike VBench2.0, which explicitly encodes physical outcomes in the prompt, VideoVerse introduces "**hidden semantics**" elements. This design forces T2V models to rely on their intrinsic world knowledge to generate implicit but necessary phenomena, enabling a more faithful evaluation of the world modeling ability.

| Comparison of Prompts in VBench2.0 and VideoVerse (Ours) | |
| --- | --- |
| **VBench2.0 (Mechanics)** | A water droplet slides down the smooth surface of a marble countertop, remaining spherical due to surface tension. |
| **VBench2.0 (Mechanics)** | A soft rubber duck is tossed onto the floor, showing its energetic bounce as it strikes the surface. |
| **VideoVerse** | As a sleek, futuristic spaceship with reflective silver plating sails across Jupiter. Then the spaceship maneuvers gracefully through a dense cluster of asteroids and dodges asteroids with precision. *Hidden Semantics:* The surface of the planet in the video has colorful striped patterns. *Evaluation (Natural Constraints):* Does the surface of the planet in the video have colorful striped patterns? |
| **VideoVerse** | An athlete wearing a red vest competes in the long jump. He sprints toward the left side of the screen and takes off, leaping high into the air and stretching his legs forward as far as possible, before landing in the sandpit on his hips. He then gets up and walks away. *Hidden Semantics:* After the athlete lands, there is a pit in the sand. *Evaluation (Interaction):* After the athlete lands, is there a pit in the sand? |

## D.3 TEMPORAL CAUSALITY OF "EVENT FOLLOWING"

Another important design of our VideoVerse is that every prompt is constructed based on at least one event. For prompts involving multiple events, we emphasize their temporal causality in most cases, aligning with our LCS-based evaluation method. As shown in Tab. 9, the three events form a strict temporal chain that cannot be reordered: if the man does not throw the frisbee, the dog cannot fetch it; if the dog does not fetch it back, the man cannot leash the dog; and once the man leashes the dog, the dog cannot fetch the frisbee. This fixed and unique order highlights the temporal causality explicitly embedded in our prompts. It is worth noting that not all prompts follow this rule; for example, in the **Science Fiction** category, we deliberately relax temporal causality to encourage model creativity.

## D.4 SCENE COVERAGE

To further assess whether VideoVerse can serve as a benchmark for evaluating the world model capabilities of T2V models, we extract the potential scenes implied by prompts like "kitchen" or "playground" in VideoVerse and several other mainstream T2V benchmarks via GPT-4o. We then measure the scene uniqueness of each benchmark. As shown in Fig. 5, VideoVerse achieves higher uniqueness by ensuring that each prompt corresponds to a distinct scene (as the 100% base). In contrast, prompts in other benchmarks are often repetitive, overly simple, or loosely related to specific scenes, resulting in lower scene uniqueness.

Table 9: An example of event-based prompt design in VideoVerse. The prompt consists of three causal events (A, B, C), which must occur in a fixed order to preserve temporal causality, consistent with our LCS-based evaluation method.

| Example of Event-Based Prompt with Temporal Causality in VideoVerse | |
| --- | --- |
| **Prompt** | A man plays fetch with a golden retriever on a grassy field. Standing on the left side of the frame, he throws a frisbee to the right, and the dog runs out to retrieve it. The man then takes out a leash and attaches it to the dog. |
| **Event A** | A man throws a frisbee. |
| **Event B** | The dog fetches the frisbee back to the man. |
| **Event C** | The man leashes his dog. |
| **Temporal Causality** | The order of events (A → B → C) is fixed and unique: If A does not occur, B cannot occur; If B does not occur, C cannot occur; Once C occurs, B cannot follow. |

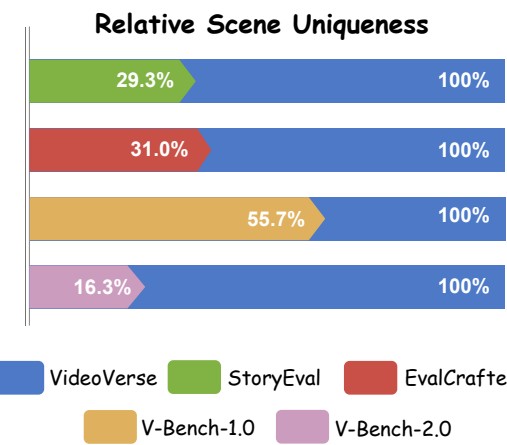

Figure 5: Comparison of scene uniqueness across T2V benchmarks. Scenes implied by prompts are extracted using GPT-4o, and semantic embeddings are used to merge similar scenes. VideoVerse achieves the highest uniqueness, ensuring broader and more diverse scene coverage.

## E   MORE CASE STUDIES

### E.1   THE GAP BETWEEN T2V MODELS AND A "WORLD MODEL"

As emphasized in Sec. 4.2 of the main paper, although current SOTA T2V models demonstrate certain abilities of a "World Model", there still exists a significant gap. We illustrate this with two cases in Fig. 6. In Fig. 6a, Minmax-Hailuo successfully generates the content of "a man shaving his beard", but the beard remains unchanged. This indicates that the model fails to capture the implicit world knowledge that "shaving a beard" also implies that "the beard should disappear", which is related to *Interaction*. In Fig. 6b, Hunyuan correctly generates dry ice and places it in the right location, but it fails to produce the vapor that should appear when dry ice is exposed to room temperature. This reveals that the model does not understand the physical knowledge that dry ice rapidly sublimates into vapor at room temperature.

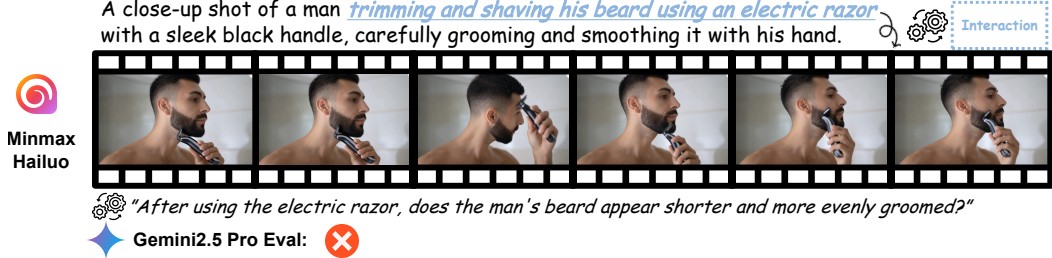

(a) Hailuo Video can generate appealing *man shaving actions*, but it fails on the dimension of **Interaction**: while the razor repeatedly moves across the beard, the beard remains unchanged.

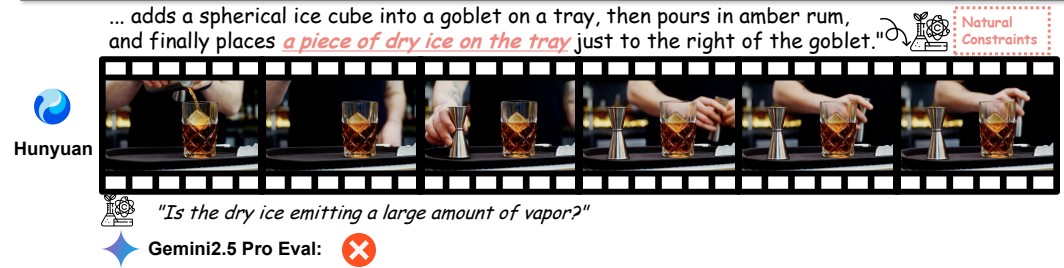

(b) Hunyuan Video can correctly generate basic visual elements such as *a spherical ice cube*, *the pouring action*, *a piece of dry ice*, and 2D layout relations like *to the right of*. However, it fails on the **Natural Constraints** dimension: the dry ice shows no sublimation at room temperature.

Figure 6: Examples illustrating the gap between current T2V models and a "world model". (a) Although Minmax-Hailuo successfully generates the action of a man shaving his beard, it fails on the dimension of *Interaction*: the beard remains unchanged despite the shaving action, indicating a lack of understanding that "shaving" implies the beard should gradually disappear. (b) Hunyuan correctly generates visual elements such as a spherical ice cube, a piece of dry ice, and their correct spatial placement. However, it fails on the *Natural Constraints* dimension: the dry ice shows no sublimation when exposed to room temperature, missing the physical knowledge that dry ice should emit vapor under these conditions.

### E.2    More Cases in VideoVerse

Fig. 8 presents more cases of VideoVerse, covering different evaluation dimensions and prompt types, including *Scientific Experiment* and *Science Fiction*. By integrating carefully designed evaluation dimensions into the prompts, VideoVerse provides a comprehensive assessment of current T2V models' capabilities from a world model perspective.

## F    Details of User Study

**Participant Selection and Answer Collection.**    Since VideoVerse aims to evaluate T2V models from a world model perspective, participants in the user study are required to have a certain level of knowledge. Following the same protocol as our data annotation process, all participants hold at least a bachelor's degree, with some at the PhD level. We provide the user interface of our user study in Fig. 7. Each participant is presented with a video without its corresponding prompt, and is asked to answer a binary question related to the video, without being informed of the associated evaluation dimension. Finally, participants are asked to order the events that occurred in the video. To ensure data quality, participants cannot submit their answers until the video is played.

**Results of User Study.**    After collecting the responses from all participants, binary questions are evaluated by directly computing the proportion of answers consistent with Gemini 2.5 Pro. For *Event Following*, we measure consistency by calculating the longest common subsequence (LCS, see Sec. 3.4 in the main paper) between Gemini's response and each participant's response, and then aggregating by multiplying Gemini's LCS score with the number of participants to yield a

Figure 7: Interface used for the user study. The participant can watch the video, but cannot see the prompts that are used to generate the video, nor do they know which model generates it. The participant is required to answer the **Basic Questions**, i.e., the binary evaluation questions and then the *Event Following* dimension. They also need to complete the **Order Selection**, arranging a series of events in order; if they believe a certain event does not occur, they can select "null".

Table 10: Detailed User Study Results.

| Evaluation Category | Consistency Ratio |
|---------------------|:-----------------:|
| Event Following     | 94.26 |
| Natural Constraints | 89.6 |
| Common Sense        | 100 |
| Interaction         | 95.5 |
| Mechanics           | 90.9 |
| Material Properties | 100 |
| Attribute Correctness | 96.2 |
| Camera Control      | 79.2 |
| 2D Layout           | 90.9 |
| 3D Depth            | 100 |
| **Overall** (Weighted) | 93.10 |

scale-adjusted consistency ratio. As shown in Tab. 10, most evaluation categories exhibit a high consistency ratio (>90%). However, the *Camera Control* exhibits the lowest consistency ratio. This is because its changes typically occur throughout the entire video, making it more difficult for the VLM to understand and requiring more granular participant observation of the video (e.g., focus control is typically only reflected in a certain area of the video). For the *Natural Constraints*, since interpreting these phenomena often requires specific scientific knowledge and domain expertise, and the underlying cues are sometimes subtle, which are difficult for the VLM. This also motivates us to require that all prompt annotators and user study participants should hold at least a bachelor's degree. For other evaluation dimensions, consistency ratios in dynamic tasks (e.g., *Mechanics* and *Camera Control*) are slightly lower than those in static ones, as answering these questions demands temporal reasoning, which is inherently more difficult. For *2D Layout*, inconsistencies mainly arise from object occlusions in the video, which can lead to confusion in spatial descriptions such as distinguishing between "left" and "right".

## G   OTHER VLM AS EVALUATOR

Although the powerful Gemini 2.5 Pro demonstrates high consistency with human evaluations, it remains a closed-source commercial model, which limits its accessibility for the community to use as an evaluator for their own T2V models. To address this issue, we explore the use of open-source VLMs as evaluators. Tab. 11 reports the evaluation results when adopting Qwen2.5-VL 32B as the evaluator (considering the balance between performance and efficiency). To assess whether it can serve as a substitute for Gemini 2.5 Pro used in our main results, we first calculate the Spearman correlation coefficient with the results in Tab. 1, obtaining a value of 0.71. This indicates that there

Table 11: Performance of Open-Source and Closed-Source T2V models on VideoVerse based on the Open-Source VLM Qwen2.5-VL 32B, with the best values highlighted in bold. The light green columns represent the world model level dimensions, while the other columns represent the basic level dimensions.

| Model | Overall | Dynamic | | | | | Static | | | | |
|---|---|---|---|---|---|---|---|---|---|---|---|
| | | Event Following | Camera Control | Interaction | Mechanics | Material Properties | Natural Constra. | Common Sense | Attr. Correct. | 2D Layout | 3D Depth |
| *Open-Source Models* | | | | | | | | | | | |
| CogVideoX1.5 (S) | 1077 | 591 | 31 | 31 | 28 | 29 | 31 | 41 | 187 | 58 | 50 |
| CogVideoX1.5 (L) | 1079 | 603 | 34 | 30 | 28 | 24 | 29 | 37 | 189 | 58 | 47 |
| SkyReels-V2 (S) | 1155 | 659 | 42 | 32 | 40 | 27 | 30 | 48 | 167 | 65 | 45 |
| SkyReels-V2 (L) | 1109 | 624 | 40 | 37 | 34 | 30 | 31 | 40 | 171 | 60 | 42 |
| Wan2.1-14B | 1152 | 632 | 47 | 28 | 41 | 29 | 30 | 45 | 183 | 67 | 50 |
| Hunyuan | 1093 | 601 | 39 | 33 | 39 | 29 | 27 | 45 | 177 | 54 | 49 |
| OpenSora2.0 | 1167 | 619 | 48 | 37 | 41 | 33 | 40 | 52 | 186 | 63 | 48 |
| Wan2.2-A14B | 1272 | 703 | 63 | 34 | 47 | 34 | 37 | 49 | 185 | 69 | 51 |
| *Closed-Source Models* | | | | | | | | | | | |
| Minimax-Hailuo | 1309 | 709 | 69 | 41 | 44 | 33 | 39 | 58 | 196 | 69 | 51 |
| Veo-3 | 1336 | 722 | 84 | 42 | 39 | 34 | 48 | 58 | 190 | 67 | 52 |

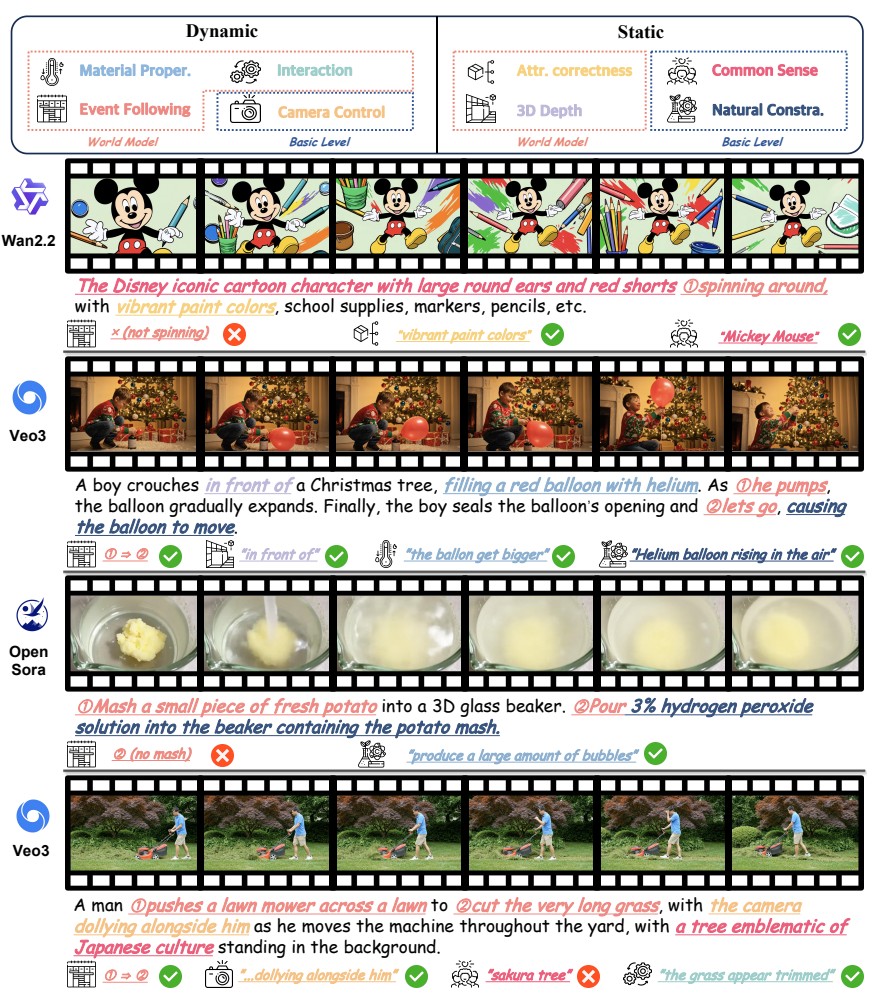

Figure 8: More examples of different T2V models in our VideoVerse.

is still a gap between the two evaluation results. The most significant differences occur in the basic static dimensions, i.e., *Attribute Correctness*, *2D Layout*, and *3D Depth*. A closer analysis reveals that for *2D Layout* and *3D Depth*, the spatial relationships between objects are often affected by

occlusion, which substantially increases the difficulty for open-source VLMs with limited video understanding ability. For *Attribute Correctness*, the discrepancy arises because the evaluated objects do not remain consistently in prominent positions throughout the video, leading to divergence from Gemini 2.5 Pro's evaluations.

For overall performance, the difference between Qwen2.5-VL 32B and Gemini 2.5 Pro is not substantial. Based on the LCS metric described in Sec. 3.4 of the main paper, the overall ranking difference between Tab. 11 (results from Qwen2.5-VL 32B) and Tab. 1 (results from Gemini 2.5 Pro) is 3. Moreover, models with changes in relative positions are already very close in performance, such as CogVideoX1.5 (L) vs. CogVideoX1.5 (S) and SkyReels-V2 (S) vs. SkyReels-V2 (L). These results suggest that current open-source VLMs can serve as evaluators for **VideoVerse**, as the overall rankings are very close without altering the conclusions of **VideoVerse**. Nonetheless, since our comprehensive user study has confirmed the high alignment of Gemini 2.5 Pro with human evaluations, and given that open-source VLMs still exhibit certain discrepancies compared to Gemini 2.5 Pro on VideoVerse, we recommend using the more powerful Gemini 2.5 Pro as the primary evaluator for VideoVerse.

## H    USAGE OF LLMS IN THIS PAPER

In this work, Large Language Models (LLMs) are employed for stylistic refinement, polishing the manuscript and processing the data. All original content and ideas are conceived and written by the authors.

