# OpenReview forum: "VideoVerse: How Far is Your T2V Generator from a World Model?"
_ICLR.cc/2026/Conference — ICLR 2026 Conference Withdrawn Submission_

### Official Review · Reviewer_d14X · 2025-10-21

**Soundness:** 2
**Presentation:** 3
**Contribution:** 3
**Rating:** 6
**Confidence:** 3

**Summary:**

This paper introduces VideoVerse, a new benchmark for evaluating text-to-video (T2V) models from a "world model" perspective. The benchmark consists of 300 carefully curated prompts with 815 events and 825 binary evaluation questions across 10 dimensions (5 dynamic, 5 static). The key innovation is the focus on "hidden semantics" - requiring models to generate content based on implicit world knowledge rather than just following explicit textual instructions. The authors evaluate several open-source and closed-source T2V models using Gemini 2.5 Pro as the evaluator and conduct a user study showing high alignment with human judgment.

**Strengths:**

- Addresses an important and timely problem. The paper clearly identifies that existing benchmarks (VBench, EvalCrafter, StoryEval) fail to capture causal and world knowledge reasoning. The emphasis on temporal causality between events and implicit physical/chemical knowledge (like dry ice sublimation, acid-base reactions) is a genuinely useful direction as these models become more sophisticated.
- Comprehensive and well-structured benchmark. (1) The paper evaluates 8 different models (both open and closed source) and provides detailed breakdowns across all 10 dimensions covering both static and dynamic reasoning. The results clearly show that even SOTA models like Veo-3 only achieve 76.4% on world model dimensions vs 81.3% on basic dimensions, which nicely demonstrates the gap. (2) The case studies effectively illustrate where models fail, making the findings concrete and interpretable. (3) I like how they connect low-level perceptual factors (camera control, 2D layout) with higher-level “world understanding” factors like natural constraints and interaction. This hierarchical setup makes the benchmark more realistic and nuanced. (4) The authors validate Gemini 2.5 Pro’s consistency with human judgments (>90% agreement), which adds credibility to the evaluation.

**Weaknesses:**

- The “world model” claim is a bit overstated. The title and intro heavily emphasize world models, but the benchmark only probes limited prompt coverage and short-duration videos (5 - 10 seconds). 300 prompts with specific evaluation dimensions can't really capture the full scope of what a world model should understand. The 10 dimensions are useful but can be further extended. There's no evaluation of prediction, simulation, social interactions, long-term planning, longer-term consistency, or many other aspects of world understanding. So it feels more like a semantic-causal benchmark than a true world-model testbed.
- Missing important baseline models and limited accessibility of evaluator. The paper doesn't include Kling, Sora, or Pika which are prominent T2V models frequently discussed in the community. The absence of Sora is particularly glaring given OpenAI's explicit positioning of it as a world simulator. While I understand there may be API access issues, this limits the completeness of the evaluation. Additionally, the reliance on Gemini 2.5 Pro (a closed-source commercial model) as the primary evaluator creates accessibility barriers. Table 11 shows Qwen2.5-VL has only 0.71 Spearman correlation with Gemini, and the authors still recommend using Gemini, which means researchers without API access can't reliably use this benchmark.
- Reliance on GPT-4o for prompt construction and annotation may introduce bias. Section 3.3 mentions using GPT-4o heavily for filtering, causality extraction, and even dimension selection. This automated pipeline could bias the dataset toward GPT-style reasoning patterns, but there’s no quantitative check of annotation diversity.

**Questions:**

Are the prompts public and reusable for benchmarking future T2V systems?

---

### Official Review · Reviewer_JioA · 2025-10-25

**Soundness:** 2
**Presentation:** 3
**Contribution:** 3
**Rating:** 4
**Confidence:** 3

**Summary:**

This paper introduces VideoVerse, a novel benchmark designed to evaluate Text-to-Video (T2V) models from the perspective of a "world model." The authors argue that existing benchmarks, which focus on frame-level quality and explicit semantic alignment, are insufficient for assessing state-of-the-art models that are beginning to demonstrate rudimentary reasoning about temporal causality and real-world knowledge. VideoVerse addresses this gap with a carefully curated set of 300 prompts that require models to infer "hidden semantics" and adhere to physical laws and common sense. The benchmark is evaluated using a VLM-based QA pipeline, and a comprehensive analysis of leading open-source and closed-source models reveals significant gaps, particularly in world-modeling capabilities.

**Strengths:**

1. The core premise of the paper is timely and important. As T2V models improve in fidelity, the field urgently needs benchmarks that push beyond superficial quality metrics and test deeper understanding. The focus on "world models" aligns with a critical long-term goal for AI.

2. The use of a powerful VLM (Gemini 2.5 Pro) for automated evaluation, combined with a well-defined cumulative scoring strategy and a specific protocol (LCS for event following, binary questions for other dimensions), is robust. The user study validating the VLM's judgments (>90% consistency) is essential for establishing the credibility of the automated evaluation method.

**Weaknesses:**

1. While the paper acknowledges that the definition of a "world model" is multi-faceted, VideoVerse primarily tests physical and commonsense reasoning. Other critical aspects of a world model, such as theory of mind, long-term planning, complex social dynamics, or counterfactual reasoning, are not addressed.

2. The entire evaluation hinges on the capabilities and biases of Gemini 2.5 Pro. Although the user study shows high alignment, it is limited in scale (15 videos, 11 participants). Potential biases in the VLM (e.g., towards certain visual concepts or narrative structures) could propagate into the benchmark scores.

3. The multi-stage pipeline involving GPT-4o and human annotators, while ensuring quality, may limit the scalability and reproducibility of the benchmark. A more detailed analysis of inter-annotator agreement during the prompt refinement stage would strengthen the methodology.

**Questions:**

See weaknesses.

---

### Official Review · Reviewer_huGy · 2025-10-28

**Soundness:** 2
**Presentation:** 3
**Contribution:** 2
**Rating:** 2
**Confidence:** 5

**Summary:**

This paper introduces VideoVerse, a new benchmark to test if T2V (Text-to-Video) models can function as "world models". It uses 300 prompts with "hidden semantics" to evaluate complex temporal causality and world knowledge, which current benchmarks miss. The study finds that all T2V models, especially open-source ones, still significantly lack these advanced reasoning abilities and are far from true world models.

**Strengths:**

1. This paper introduces VideoVerse, the first benchmark designed to specifically evaluate T2V models from a "world model" perspective. This fills a critical gap, as existing benchmarks are insufficient for assessing complex temporal causality and world knowledge.
2. The benchmark employs a novel "hidden semantics" prompt design. This requires models to infer and generate realistic outcomes based on physical laws or common sense (e.g., "baking soda and vinegar" should produce "bubbles") , testing deeper reasoning rather than just explicit instruction-following.

**Weaknesses:**

1. The definition of "world model" in this paper is not clear. It seems "world model" here refers to T2V models that can work well on a set of pre-defined dimensions. However, a world model is expected to face an "open world," and it is questionable whether its capabilities can be adequately measured by a small number of fixed dimensions.

2. The benchmark is predicated on the assumption that Gemini 2.5 Pro (the evaluator) is already a "world model" that understands the world. However, the authors provide no literature to support this claim. Furthermore, the user study was conducted using only 15 videos, and an experiment with such a small dataset is unconvincing.

3. The number of prompts is scarce. Although the authors emphasize that the 300 prompts are carefully designed, the scope of "world knowledge" and "natural laws" is extremely vast. These 300 prompts likely represent only a very small subset of this knowledge, which limits the complexity of the natural constraints being tested.

**Questions:**

See weakness.

---

### Note · Authors · 2025-11-12

**Comment:**

We are deeply grateful to all the reviewers for their valuable time. We express our gratitude for the valuable suggestions of the reviewers. I have read and agree with the venue's withdrawal policy on behalf of myself and my co-authors.

**Withdrawal Confirmation:**

I have read and agree with the venue's withdrawal policy on behalf of myself and my co-authors.